# Engineering of Genetically Encoded Bright Near-Infrared Fluorescent Voltage Indicator

**DOI:** 10.3390/ijms26041442

**Published:** 2025-02-08

**Authors:** Xian Xiao, Aimei Yang, Hanbin Zhang, Demian Park, Yangdong Wang, Balint Szabo, Edward S. Boyden, Kiryl D. Piatkevich

**Affiliations:** 1School of Life Sciences, Westlake University, Hangzhou 310024, China; xiaoxian@westlake.edu.cn (X.X.); zhanghanbin@westlake.edu.cn (H.Z.); wangyangdong76@gmail.com (Y.W.); 2Westlake Laboratory of Life Sciences and Biomedicine, Hangzhou 310024, China; 3Institute of Basic Medical Sciences, Westlake Institute for Advanced Study, Hangzhou 310024, China; 4McGovern Institute for Brain Research, Massachusetts Institute of Technology (MIT), Cambridge, MA 02139, USA; amyang@mit.edu (A.Y.); demian@media.mit.edu (D.P.); 5CellSorter KFT, H-1117 Budapest, Hungary; info@cellsorter-scientific.com; 6Department of Biological Physics, Eötvös Loránd University (ELTE), H-1053 Budapest, Hungary; 7Howard Hughes Medical Institute, Cambridge, MA 01239, USA; 8Center for Neurobiological Engineering, K. Lisa Yang Center for Bionics, MIT, Cambridge, MA 01239, USA; 9Department of Media Arts and Sciences, MIT, Cambridge, MA 01239, USA; 10Department of Brain and Cognitive Sciences, MIT, Cambridge, MA 01239, USA; 11Department of Biological Engineering, MIT, Cambridge, MA 01239, USA; 12Koch Institute, MIT, Cambridge, MA 01239, USA

**Keywords:** genetically encoded voltage indicator, rhodopsin, brightness, near infrared, directed molecular evolution, phototoxicity

## Abstract

Genetically encoded voltage indicators (GEVIs) allow for the cell-type-specific real-time imaging of neuronal membrane potential dynamics, which is essential to understanding neuronal information processing at both cellular and circuit levels. Among GEVIs, near-infrared-shifted GEVIs offer faster kinetics, better tissue penetration, and compatibility with optogenetic tools, enabling all-optical electrophysiology in complex biological contexts. In our previous work, we employed the directed molecular evolution of microbial rhodopsin Archaerhodopsin-3 (Arch-3) in mammalian cells to develop a voltage sensor called Archon1. Archon1 demonstrated excellent membrane localization, signal-to-noise ratio (SNR), sensitivity, kinetics, and photostability, and full compatibility with optogenetic tools. However, Archon1 suffers from low brightness and requires high illumination intensities, which leads to tissue heating and phototoxicity during prolonged imaging. In this study, we aim to improve the brightness of this voltage sensor. We performed random mutation on a bright Archon derivative and identified a novel variant, monArch, which exhibits satisfactory voltage sensitivity (4~5% ΔF/F_AP_) and a 9-fold increase in basal brightness compared with Archon1. However, it is hindered by suboptimal membrane localization and compromised voltage sensitivity. These challenges underscore the need for continued optimization to achieve an optimal balance of brightness, stability, and functionality in rhodopsin-based voltage sensors.

## 1. Introduction

The real-time recording of membrane potential dynamics in individual neurons and neural circuits has been a long-standing goal in neuroscience research. It underpins virtually all aspects of brain function, from sensory processing to learning and memory. However, the sub-millisecond scale of neuronal membrane potential dynamics poses high demands on voltage sensors, requiring them to possess exceptional brightness, fast kinetics, and precise membrane localization [1]. As a result, the implementation of voltage indicators in neuroscience research has lagged behind the widespread use of calcium indicators, which have been more readily optimized for high-performance imaging [2,3,4]. Recently, improved genetically encoded voltage indicators (GEVIs) and advances in optical imaging instrumentation greatly facilitated the use of voltage imaging techniques in investigating brain function from the cellular level to behavioral analysis [5].

GEVIs are primarily classified into three main families based on their voltage-sensing mechanisms: voltage-sensing domain (VSD)-based indicators, microbial rhodopsin-based indicators, and chemogenetic or hybrid probes [5,6]. Among these, near-infrared voltage indicators derived from rhodopsins offer several advantages, including high temporal fidelity, reduced light scattering in brain tissue, and compatibility with blue-shifted optogenetic tools [7], thus enabling all-optical electrophysiology [8,9]. Over a decade ago, Adam Cohen’s group discovered that a microbial rhodopsin, Archaerhodopsin-3 (Arch), a light-driven proton pump, possesses dim near-infrared fluorescence and that its fluorescence increases with the depolarizing membrane voltage [10]. Since then, a series of Arch-based voltage indicators have been developed [11,12,13,14,15]. The voltage sensitivity of Arch and Arch-based voltage sensors involves the protonation of the Schiff base in the retinal chromophore [16,17]. In our previous work, we employed directed molecular evolution of Archaerhodopsin-3 in mammalian cells to develop a high-performance GEVI named Archon1 [11]. Archon1 exhibited excellent membrane localization, signal-to-noise ratio (SNR) per single action potential (AP), sensitivity, kinetics, photostability, and full compatibility with optogenetic control [11] enabling the robust voltage imaging of neuronal populations in behaving mice [18,19]. Despite being several times brighter than its precursors, Archon1 still faced a critical limitation: it remained relatively dim compared with green GEVI counterparts and required very high illumination intensities (~1.5 W/mm^2^), leading to potential issues such as heating and phototoxicity upon extended imaging.

By using Archon1 as a starting template, Cohen’s group implemented pooled video-based screening and photoselection to further evolve the variant, producing two improved Archaerhodopsin-derived GEVIs: QuasAr6a and QuasAr6b [12]. These indicators demonstrated modest improvements in per-molecule brightness compared with Archon1, with QuasAr6a and QuasAr6b achieving 1.4-fold and 1.7-fold increases, respectively. However, these enhanced variants did not achieve the dramatic brightness improvement necessary to reduce illumination power for in vivo imaging, prompting us to explore alternative templates to improve rhodopsin-based sensors further.

In the course of Archon1 development, we identified a variant, referred to as Archon3, that exhibited a five-fold increase in baseline brightness compared with Archon1 (see variant#3 in Supplementary Figure S10 in ref. [11]). By leveraging Archon3 as a starting template, we pursued directed molecular evolution to engineer brighter and more sensitive voltage indicators. By screening large gene libraries expressed in HEK cells by using automated microscopy-based screening, we identified a novel variant, named mammalian-optimized new Arch, or monArch for short, with enhanced brightness. The monArch indicator was characterized in neurons in primary culture and in acute brain slices and compared with its precursor Archon3. In addition, we created a soma-targeted version of monArch and compared it with somArchon. While monArch displayed satisfactory voltage sensitivity (4~5% ΔF/F_AP_), it was hindered by its suboptimal membrane localization. These challenges underscore the need for continued optimization to achieve an optimal balance of brightness, stability, and functionality in rhodopsin-based voltage sensors.

## 2. Results

### 2.1. Directed Evolution of Voltage Sensor Archon3 in HEK Cells

To develop a brighter sensor, we employed directed molecular evolution in HEK cells following a previously reported approach (ref. [11]) and using Archon3 as the starting template (Figure 1a). Correspondingly, we generated a gene library with random mutations throughout the opsin gene by using error-prone PCR and cloned it into the mammalian expression vector. The library, containing up to ~10^7^ independent clones, was then introduced into HEK293FT cells by using the modified calcium phosphate transfection protocol, which was optimized to deliver one plasmid per transfected cell. After 48 hours’ expression, positive cells with fluorescence excited by a 640 nm laser were sorted out by using FACS and recovered in the culture dish. After the 24 h recovery period, the cells were imaged under the automated fluorescence microscope equipped with a robotic cell picker. Cells displaying good fluorescence brightness and proper membrane localization were isolated and subjected to DNA amplification to recover target genes. The retrieved target genes were then cloned into the pN1 vector and transfected into HEK cells individually to assess voltage responses under induced transmembrane voltage (ITV) stimulation. In this way, we obtained seven variants, which were selected for further assessment in cultured neurons (Figure 1a).

Subsequently, we expressed the selected variants in primary cultured neurons and assessed their membrane localization, baseline brightness, and functional response (Figure 1b–d). Among them, three variants demonstrated detectable optical fluorescence alterations in response to spontaneous neuronal activity (Figure 1c). Variant#3 stood out with the highest ΔF/F and SNR per single AP (Figure 1d), and it also exhibited higher photostability when compared with the other two functional variants (Figure 1e). The selected variant, named monArch, contained eight mutations compared with Archon3 (Appendix A).

Next, we explored the structural mechanism underlying the increase in brightness of monArch. By using AlphaFold3, we generated predicted structure models for Arch-derived voltage indicators, namely, Archer1, QuasAr1, QuasAr2, Archon1, Archon2, Archon3, QuasAr6a, and QuasAr6b. Although these voltage indicators contain multiple mutations scattered across the protein, the predicted structures showed good convergence (Appendix A), with a root-mean-square deviation (RMSD) of less than 0.21 Å compared with Archon1. This suggests that the global structure of monArch is not significantly altered after mutations; subtle changes in the local environment underlie the brightness change. We then mapped each residue that was different from Archon1 onto the predicted monArch structural model (Figure 1f). We found that the mutations were scattered throughout the scaffold, both inside and on the surface of the protein (Figure 1f, Appendix A). Of note, we noticed that two mutations, T99C and A225G, in Archon3/monArch were in close proximity to the chromophore (within 4.0 Å; Appendix A), which may affect the chemical environment of the Schiff base, thus affecting the basal brightness and voltage sensing. Taken together, by using directed molecular evolution, we identified a brighter voltage sensor variant, monArch, with multiple mutations throughout the scaffold.

### 2.2. Characterization of monArch in Primary Cultured Hippocampal Neurons and Mouse Brain Slices

To evaluate the outcome of the mutations introduced during directed evolution, we carried out a side-by-side comparison of Archon3 and monArch in cultured neurons. To express the indicators in neurons, we fused them with Golgi export trafficking signal (KGC) and endoplasmic reticulum export sequences (ER2) to improve their membrane trafficking and EGFP tag to facilitate searching for positive neurons in the GFP channel as was previously applied for Archon1 (Figure 2a). To account for variations in expression levels, we normalized the voltage sensor fluorescence intensity to that of EGFP. We found that the red-to-green ratio for monArch is about 1.9-fold higher than its parental template Archon3 (Figure 2b; *p* = 0.03, two-tailed Student’s *t*-test). We then recorded the voltage sensor fluorescence in response to the evoked action potential via a whole-cell patch clamp. Both Archon3 and monArch were sensitive enough to report individual action potential evoked with a series of 2 ms current pulses (Figure 2c,d). Taken together, these data showed that monArch could respond to the evoked action potentials.

We also expressed monArch in cortical pyramidal neurons under the CAG promoter via in utero electroporation. After 4 weeks of expression in vivo, we prepared acute brain slices and performed voltage imaging in brain slices. We observed fast monArch fluorescence changes with an SNR of 4.3, reflecting the spontaneous activity of neurons (Figure 3). In a subset of neurons both in culture and in brain slices, we observed puncta inside the monArch-expressing neurons with somewhat compromised membrane expression.

### 2.3. Engineering of Soma-Targeted monArch

During in vivo imaging, the neuropil signal often contributes to the background, making it difficult to accurately interpret fluorescence signals. To minimize neuropil contamination in dense brain tissue, we developed a soma-targeted version of monArch by attaching a Kv2.1 trafficking motif to the C terminus of monArch7 [12,13,20,21]. In addition, we also inserted the Pro-Pro (PP) sequence after monArch, as it was shown that the PP sequence could enhance the SNR for paQuasAr3 [13]. The new molecule, monArch-PP-KGC-EGFP-Kv2.1motif-ER2, soma-monArch for short, exhibited good soma-localization (Figure 4a). The fluorescence intensity was slightly reduced compared with monArch; however, it was still 9-fold brighter than somArchon (Figure 4b,c). We also measured the photobleaching curves for soma-monArch, and we found that after 60 s of continuous imaging, somArchon and soma-monArch retained 93% and 89% of fluorescence, respectively, suggesting that soma-monArch exhibited similar photostability to somArchon. As for voltage sensitivity, the fluorescence changes per AP for monArch is about one-third of that for somArchon (Figure 4e,f).

We then assessed the cellular localization of soma-monArch in the mouse brain through histological analysis (Appendix A). We found that soma-monArch was predominantly localized to the soma and proximal dendrites (Appendix A), confirming its intended localization. However, we also observed a significant fraction of monArch trapped inside the cell, suggesting compromised membrane trafficking, which hindered further assessment of its potential for voltage imaging in brain slices and in vivo.

## 3. Discussion

In this study, we performed directed molecular evolution of the brightest Arch derivative in HEK 293FT cells and identified a brighter variant, monArch, which was characterized by 9-fold brighter baseline fluorescence compared with Archon1. monArch effectively responded to APs in mouse brain neurons, demonstrating its potential for voltage imaging applications.

Brighter voltage indicators are essential to the high-quality imaging of neuronal activity, reducing phototoxicity and enabling more accurate, sensitive, and efficient detection of voltage changes in complex biological systems. Unlike calcium signaling, which typically operates on a temporal scale of tens of milliseconds to several seconds [22,23], membrane potential responses occur on a millisecond-to-sub-millisecond timescale during neuronal activity, requiring high frame rates for capturing events with 1–2 ms durations [24]. At such high sampling rates, the number of photons detected by scientific-grade high-speed cameras is still quite limited. Therefore, to achieve a sufficient photon budget at kHz sampling frequencies, voltage indicators must be bright enough to provide reliable signals. This poses greater challenges to voltage sensor development than other sensors. In this study, we developed a new bright voltage indicator, monArch. We employed high light intensity excitation at 637 nm during imaging to ensure a consistent and accurate comparison of brightness and voltage response. Of note, it is also feasible to image Archon/monArch by using 568 nm light with lower light intensities, as Milosevic et al. used in their study [25]. This wavelength is close to the peak absorption according to the spectrum of Arch-based GEVIs [14,17,26]. Overall, despite its suboptimal membrane localization and compromised voltage sensitivity, the 9-fold increase in basal fluorescence intensity makes monArch a promising starting point for the development of next-generation rhodopsin-based voltage sensors.

Based on the structural modeling of monArch, we found that the distribution of mutation was scattered throughout the protein, which is also in agreement with previous studies showing that beneficial mutations were distributed throughout the Arch scaffold [10,20], including the protein surface. These surface mutations in monArch may influence membrane trafficking as well 13, potentially impacting the sensor’s overall performance. In addition, we also noticed that two amino acids (Cys99 and Gly225), which were in close proximity (within 4 Å) to the chromophore, were mutated in monArch and Archon3, compared with Archon1, which might account for the improved brightness compared with Archon1. In particular, the Q95E and T99C substitutions in Archon3/monArch are also present in Archer1 (Supplementary Figure S1 of [15]). By using a combination of spectroscopic studies and molecular dynamics simulations, Silapetere et al. have identified that the interaction between Q95E and T99C affects hydrogen bond network rearrangement around the chromophore, and these two sites are critical for basal brightness and voltage sensing [26]. Overall, given that the voltage-sensing mechanism of Arch-derived sensors is not yet fully understood, there are no clear consensus hotspots for structure-guided design. However, with the increasing number of Arch variants, machine learning-guided design will likely play a key role in the future development of far-red GEVIs.

## 4. Methods

### 4.1. Molecular Cloning and Mutagenesis

Synthetic DNA oligonucleotides for cloning were obtained from Integrated DNA Technologies (Coralville, ID, USA) or Tsingke Biotechnology (Beijing, China). PrimeStar Max mastermix (Takara, Osaka, Japan) was used for high-fidelity PCR amplification. Restriction endonucleases (New England BioLabs, Ipswich, MA, USA) were used following the manufacturer’s guidelines. Ligation reactions were performed with T4 DNA ligase (Fermentas, Burlington, VT, USA) or InFusion HD kits (Clontech, Mountain View, CA, USA). Plasmid DNA was isolated on a small scale by using Mini-Prep kits (Qiagen, Hilden, Germany), while large-scale plasmid purification was conducted by using GenElute HP Endotoxin-Free Plasmid Maxiprep Kits (Sigma-Aldrich, Saint Louis, MO, USA) or a Plasmid Plus Midi Kit (Qiagen, Hilden, Germany). Random mutagenesis was performed with GeneMorph II Random Mutagenesis Kits (Stratagene, La Jolla, CA, USA), under conditions yielding a mutation frequency of up to 15 mutations per 1000 base pairs. Archon3 random libraries in the pN1 vector (Clontech, Mountain View, CA, USA) were electroporated into NEB10-beta E. coli host cells (New England BioLabs, Ipswich, MA, USA). To assess electroporation efficiency, serial dilutions (10^−4^ and 10^−5^) of the electroporated cells were cultured on LB/agar plates (100 µg/mL kanamycin). The remainder cells were cultured overnight in LB medium for subsequent plasmid DNA extraction.

To express voltage sensors in the cultured hippocampal neurons, the selected Archon3 variants were PCR-amplified and swapped with the Archon1 gene in pAAV-CaMKII-SomArchon-KGC-GFP-ER2 (Addgene plasmid #126942) by using BamHI/AgeI sites.

All plasmids generated in this study are available at the WeKwikGene plasmid repository (https://wekwikgene.wllsb.edu.cn/, accessed on 4 February 2025): pAAV-CaMKII-monArch-KGC-GFP-ER2 (#0001082, https://wekwikgene.wllsb.edu.cn/plasmids/0001082, accessed on 7 February 2025); pAAV-CAG-monArch-PP-KGC-GFP-Kv2.1-ER2 (#0001083; https://wekwikgene.wllsb.edu.cn/plasmids/0001083, accessed on 4 February 2025); pAAV-CaMKII-monArch-PP-KGC-GFP-Kv2.1-ER2 (#0001084; https://wekwikgene.wllsb.edu.cn/plasmids/0001084, accessed on 4 February 2025).

### 4.2. HEK Cell Culture and Transfection

HEK293FT cells (Invitrogen, Waltham, MA, USA) were cultured in DMEM medium (Cellgro, Lincoln, NE, USA) supplemented with 10% heat-inactivated FBS (Corning, New York, NY, USA), 1% penicillin/streptomycin (Cellgro, Lincoln, NE, USA), and 1% sodium pyruvate (BioWhittaker, USA) and incubated in a 95% air/5% CO_2_ cell culture incubator, with passaging every 3–4 days. When the cell density reached approximately 70%, HEK293FT cells were transfected with the Archon3 gene library by using a commercially available calcium phosphate (CaPhos) transfection kit (Life Technologies, Carlsbad, CA, USA) as previously described 10. To ensure that only one copy of the Archon3 variant plasmid was transfected in each cell, the gene library was transfected in a 1:100 ratio with the empty vector pUC19 as “dummy” DNA to achieve low-copy transfection of the Archon3 variants.

### 4.3. FACS Screening and Single-Cell Isolation Using a Cell Picker

HEK293FT cells transfected with the gene library were harvested 48 h post-transfection. Cells were treated with trypsin (Cellgro, Lincoln, NE, USA) for 5–10 min, followed by two washes with PBS (Cellgro, Lincoln, NE, USA) by centrifuging the cell suspension for 5 min at 500 rpm at 4 °C to remove the trypsin. After the last centrifugation, cells were resuspended in PBS containing 4% FBS (Corning, New York, NY, USA) and 10 mM EDTA at a density of 1–2 × 10^6^ cells/mL. To prevent clogging during sorting, the cell suspension was filtered through a 30 µm filter (Falcon, Doral, FL, USA). The filtered cells were sorted by using a FACSAria flow cytometer (BD Biosciences) equipped with 405-, 488-, 561-, and 640-nm solid-state lasers, and data were acquired with BD FACS Diva8.0 software. Debris, dead cells, and aggregates were excluded via forward and side scatter gating. Fluorescence was detected at 640 nm excitation and 710/50 nm emission. Cells with higher fluorescence intensity than the positive control (HEK293FT cells transfected with the template protein) were sorted and collected into 5 mL tubes. The sorted cells were subsequently plated onto 3 cm dishes pre-coated with Matrigel (BD Biosciences, San Jose, CA, USA). Approximately 80% of the sorted cells were positive for fluorescence, as confirmed by wide-field fluorescence imaging.

After a 24 h recovery period in a culture dish, the sorted cells were washed to remove non-attached cells and subjected to microscope-guided cell screening by using our single-cell manipulation system as previously described [11]. Fluorescent images of cells in the culture dish were acquired and analyzed by using custom MATLAB 2019b code to quantify the brightness (mean fluorescence signal) of every detected single cell. The cells with the highest brightness were picked into PCR tubes and stored at −80 °C for target gene recovery.

### 4.4. Target Gene Recovery

Cells collected in individual PCR tubes underwent whole-genome amplification (REPLI-g WGA kit, Qiagen, Hilden, Germany), followed by PCR amplification. The resulting amplicons were subjected to agarose gel electrophoresis. The bands with the correct size were purified and cloned into an expression vector. The purified plasmids were individually transfected into HEK cells for the expression and characterization of each gene. To account for the potential mutagenic effects of HEK293T cells on exogenous DNA, as well as the possibility of multiple plasmids being incorporated into a single cell, at least five different genes were screened from each isolated HEK cell.

### 4.5. Induced Transmembrane Voltage (ITV) in HEK Cells

To assess voltage sensitivity, HEK293FT cells that express specific mutants were exposed to an electric field generated by a pair of platinum electrodes as described previously described 10. HEK293FT cells were cultured in 24-well plates, each receiving 500 ng of the target plasmid DNA for transfection, following the protocol outlined by TransIT-X2 (Mirus Bio, Madison, WI, USA). For cell imaging, we utilized an inverted Eclipse Ti-E microscope from Nikon (Tokyo, Japan), which was equipped with a CMOS camera (Zyla5.5, Andor, London, UK), LEDs (Spectra, Lumencor, Beaverton, OR, USA), and a 637 nm laser (637 LX, OBIS, Santa Clara, CA, USA) focused on the back focal plane of a 40× objective with an NA of 1.15 (Nikon, Tokyo, Japan). The imaging system also included a filter set with a 664LP filter for long-pass emission and a 650 nm dichroic filter (Semrock, Rochester, NY, USA). The platinum electrodes, spaced 4 mm apart and attached to our automated micromanipulator, were systematically positioned across the wells. Electrical pulses, ranging from 20 to 100 V/cm, with a duration of 50 ms and a frequency of 2 Hz, were generated by a DG2041A Arbitrary Waveform Function Generator (RIGOL, Gilching, Germany) and then amplified by using a high-voltage amplifier (Model 2205, Trek, Lockport, NY, USA). These pulses were used to stimulate changes in the cell membrane voltage. Fluorescent images were captured at a rate of 200 frames per second in 2 × 2 binning mode for 20 s.

### 4.6. Primary Hippocampal Neuronal Culture and Transfection

All animal procedures at MIT were performed in compliance with the US National Institutes of Health Guide for the Care and Use of Laboratory Animals and were approved by the Massachusetts Institute of Technology Committee on Animal Care. Hippocampal neurons were isolated from postnatal day 0 or 1 Swiss Webster mice (regardless of sex) as described previously 10. Hippocampal tissue was dissected and digested with 100 units of papain (Worthington Biochem, Lakewood, NJ, USA) for 6–8 min. Digestion was terminated by adding an ovomucoid trypsin inhibitor (Worthington Biochem, USA). The cells were plated at a density of 20,000–30,000 per glass coverslip coated with Matrigel (BD Biosciences). Neurons were seeded in 100 µL plating medium consisting of MEM (Life Technologies, Carlsbad, CA, USA), glucose (33 mM; Sigma, Ronkonkoma, NY, USA), transferrin (0.01%; Sigma, Ronkonkoma, NY, USA), Hepes (10 mM; Sigma, Ronkonkoma, NY, USA), Glutagro (2 mM; Corning, New York, NY, USA), insulin (0.13%; St. Louis, Millipore, MO, USA), B27 supplement (2%; Gibco, Waltham, MA, USA), and heat-inactivated FBS (7.5%; Corning, New York, NY, USA). After the cells adhered, additional plating medium was added. AraC (0.002 mM; Sigma, Ronkonkoma, NY, USA) was introduced once the glial cell density reached 50–70% confluence.

For obtaining the results presented in Figure 4, the neurons were isolated from postnatal day 0 C57BL6 mice. The dissected hippocampal tissues were digested with 0.25% trypsin (Gibco, 25200056, Waltham, MA, USA) for 12 min at 37 °C, with gentle shaking every 4 min. The digestion was terminated by using a complete medium containing DMEM supplemented with 10% FBS preheated at 37 °C. The digested tissue was then gently dissociated and filtered through a 40 μm nylon strainer. The dissociated neurons were plated at a density of 150,000 per well on pre-coated coverslips (ThermoFisher Scientific™, 1254580, Waltham, MA, USA) with Matrigel (BD Biosciences, 356235, San Jose, CA, USA). After 12 h of incubation at 37 °C/5% CO_2_ for cell adhesion, half of the medium was replaced by Neurobasal medium (GIBCO, 21103049) supplemented with 1% GlutaMax (GIBCO, 35050061, Waltham, MA, USA) and 2% B27 (GIBCO, 17504044, Waltham, MA, USA).

Neurons cultured in vitro were transfected at 4–5 days in vitro (DIV) by using a commercial calcium phosphate transfection kit (Life Technologies, Carlsbad, CA USA), as previously described [10]. Briefly, 500 ng of plasmid DNA per well was used for transfection, followed by an additional wash with acidic MEM buffer (pH 6.7–6.8) after 30–60 min of incubation with the calcium phosphate precipitate to remove any residual precipitates [27]. All measurements on the neurons were taken between DIV 14 and DIV 18 (~9–14 days post-transfection) to allow for sufficient time for sodium channel maturation.

### 4.7. Fluorescence Microscopy of Primary Neurons

For obtaining the results in Figure 1 and Figure 2, the fluorescent imaging of voltage sensors expressed in cultured hippocampal neurons was conducted by using a Nikon Eclipse Ti inverted microscope. The system was equipped with a 40× NA 1.15 water immersion objective (Nikon, Tokyo, Japan), a 637 nm laser (637 LX, OBIS, Santa Clara, CA, USA) focused on the back focal plane of the objective, and a SPECTRA- X light engine (Lumencor, Beaverton, OR, USA) with excitation filters of 475/28 nm, 585/29 nm, and 631/28 nm (Semrock, Rochester, NY, USA). A 470 nm LED (Thorlabs, Newton, NJ, USA) and a 5.5 Zyla camera (Andor, UK) were used for imaging, all controlled by NIS-Elements AR software (versions 5.21.00 and 5.30.00).

For obtaining the results in Figure 4, cultured primary neurons were imaged with a Nikon Ti2-E widefield microscope equipped with Spectra III Light Engine (LumenCore, Beaverton, OR, USA), the ORCA-Flash 4.0 V3 sCMOS camera (Hamamatsu, Shizuoka, Japan), and a 40×/1.15 NA water immersion objective controlled by NIS Elements (AR 5.21.00). The voltage sensors and GFP were imaged by using the red channel (637 nm excitation laser, emission at 664 nm long pass) and green (excitation at 475/28 nm and emission at 535/46 nm), respectively.

### 4.8. AAV Injection and Histology

All animal care and experimental procedures involving mice were carried out in accordance with the animal care guidelines of Westlake University, and all studies were approved by the Institutional Animal Care and Use Committee (IACUC) of Westlake University under protocol No. 19-044-KP-2. C57BL/6 mice, obtained from the Animal Facility at Westlake University, were used regardless of sex. All mice were housed either in family groups or in pairs in a temperature-controlled environment with a 12 h light/dark cycle. Mouse pups at postnatal days P0-P1 received injections of rAAV2/9-CaMKII-monArch mutants-PP-KGC-GFP-Kv2.1-ER2. The viral preparations, with a titer greater than 10^12^ viral genomes per milliliter, were sourced from Shanghai Sunbio Medical Biotechnology and were administered pan-cortically by using a Hamilton microliter syringe. Before the procedure, the pups were cooled on ice for 3–5 min to achieve hypothermic anesthesia. A volume of 0.4–0.6 µL of the virus solution, which included 10% FastGreen dye (SigmaAldrich, Saint Louis, MO, USA), was manually injected into the cortical region. After that, the pups were placed on a heating pad to recuperate and then returned to their home cage. Approximately one month after AAV injection, the mice were anesthetized with an intraperitoneal injection of pentobarbital (100–150 mg/Kg) and transcardially perfused with PBS followed by 4% PFA (Beyotime, Shanghai, China). Brains were post-fixed overnight in 4% PFA. Coronal brain slices were prepared by using a vibrating microtome (Leica VT1000S, Nussloch, Germany) with 50 µm thickness. After that, slices were washed with PBS three times and incubated with DAPI (1:2000 dilution in PBS; Abcam, Cat#ab228549-2m, Waltham, MA, USA) and mounted with ProLong™ Gold Antifade (Thermo Fisher, Cat# P36930, St. Bend, OR, USA).

### 4.9. In Utero Electroporation

Female Swiss Webster (Taconic Biosciences, Albany, NY, USA) mice at embryonic day 15.5 were deeply anesthetized by using 2% isoflurane. The uterine horns were then exposed and regularly washed with warm sterile phosphate-buffered saline (PBS). The pAAV-CAG-monArch-KGC-EGFP-ER2-WPRE plasmid (1 μg/μL diluted in PBS) was injected into the lateral ventricle of one cerebral hemisphere of each embryo. This was followed by the application of five voltage pulses (50 V, 50 ms duration, 1 Hz) by using round plate electrodes with the aid of electroporators (CUY21 from NEPA GENE; ECM 830 from Harvard Apparatus, Holliston, MA, USA). After injection and electroporation, the embryos were returned to their mothers. All procedures were conducted in compliance with the protocols approved by the Massachusetts Institute of Technology Committee on Animal Care.

### 4.10. Voltage Imaging in Acute Brain Slices

Brain slices were prepared from electroporated mice without regard to sex at postnatal day (P)12–P22. Mice were anesthetized by isoflurane inhalation and decapitated, and cerebral hemispheres were quickly removed and placed in a choline-based cutting solution (in mM): 110 choline chloride, 25 NaHCO_3_, 2.5 KCl, 7 MgCl_2_, 0.5 CaCl_2_, 1.25 NaH_2_PO_4_, 25 glucose, 11.6 ascorbic acids, and 3.1 pyruvic acids. Coronal slices (300 μm thick), were sectioned by using a Leica VT1000s vibratome, then transferred to a holding chamber with cold artificial cerebrospinal fluid (ACSF) containing 125 NaCl, 2.5 KCl, 25 NaHCO_3_, 2 CaCl_2_, 1 MgCl_2_, 1.25 NaH_2_PO_4_, and 11 glucose, and allowed to recover for 30 min at 34 °C. The slices were then kept at room temperature for at least 30 min until they were ready for use. Both the cutting solution and ACSF were continuously oxygenated with a 95% O_2_/5% CO_2_ mixture.

Optical recordings from the acute brain slices were acquired with a conventional one-photon fluorescence microscope equipped with an ORCA Flash 4.0 V3 Digital CMOS camera (Hamamatsu Photonics K.K., C13440-20CU, Shizuoka, Japan), a 10× NA0.25 LMPlanFI air objective (Olympus, Tokyo, Japan), a 40× NA0.8 LUMPlanFI/IR water immersion objective (Olympus, Tokyo, Japan), a 470 nm LED (Thorlabs, M470L3, Newton, NJ, USA), a 140 mW 637 nm red laser (Coherent, Obis 637-140X, Saxonburg, PA, USA), a green filter set with a 470/25 nm bandpass excitation filter, a 495 nm dichroic, a 525/50 nm bandpass emission filter, and a near-infrared filter set with a 635 nm laser dichroic filter and a 664 nm long-pass emission filter.

### 4.11. Data Analysis

Data were analyzed offline by using NIS-Elements Advance Research software (versions 5.21.00 and 5.30.00), Origin (OriginPro 2019b, OriginLab, Northampton, MA, USA), Excel2016 (Microsoft), and Fiji ImageJ 2.9.01/1.53t. For fluorescence imaging data, regions of interest (ROIs) encompassing individual cell bodies and adjacent cell-free areas (used as background) were manually selected. Fluorescence traces from Archons or monArch were corrected for photobleaching by subtracting the baseline fluorescence. Fluorescence changes were quantified by using the formula ΔF/F_AP_ = (Fpeak − Fbl)/Fbl, where Fpeak represents the maximum fluorescence intensity during an action potential (AP), Fbl (baseline fluorescence) is the mean fluorescence intensity averaged over the 100 to 200 ms before an AP. The SNR for an AP was calculated by dividing the peak fluorescence of an AP by the standard deviation (S.D.) of the baseline fluorescence over a 100 to 200 ms window preceding the AP. The SNRs obtained during a recording session were averaged to determine the AP SNR for each cell.

## Figures and Tables

**Figure 1 ijms-26-01442-f001:**
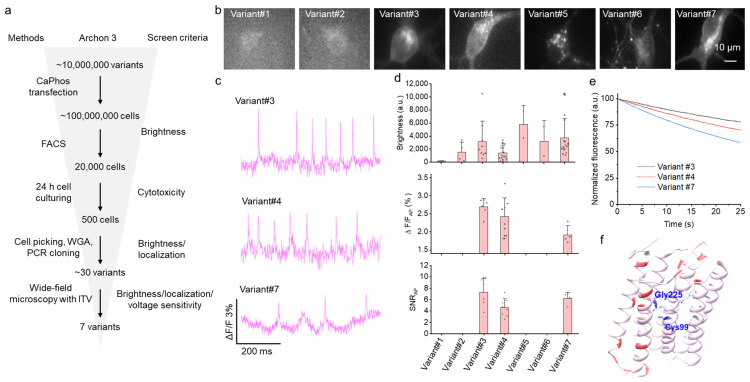
Multiparameter-directed evolution of voltage sensor Archon3 in mammalian cells. (**a**) Screening workflow for multiparameter optimization of genetically encoded voltage sensors based on Archon3 in HEK293FT cells. CaPhos, calcium phosphate transfection. (**b**) Expression of the selected 7 variants in cultured hippocampal neurons. (**c**) Representative fluorescence optical traces showing spontaneous activity of neurons expressing different voltage sensor variants. (**d**) Brightness, ΔF/F, and SNR for the selected 7 variants. Data are presented as means ± S.D. *n* = 1~15 neurons from 1~3 cultures. (**e**) Photobleaching curves of variants #3, #4, and #7 under continuous illumination (*n* = 9, 8, and 15 neurons from 1~3 cultures, respectively). λem = 664 long pass at 1.5 W/mm^2^. (**f**) Structural model of variant #3 (monArch) as predicted by AlphaFold3. The chromophore is colored in cyan. The mutation sites are highlighted in red (compared with Archon1). The two mutation residues Gly225 and Cys99, which are within 4 Å from the chromophore, are highlighted in blue.

**Figure 2 ijms-26-01442-f002:**
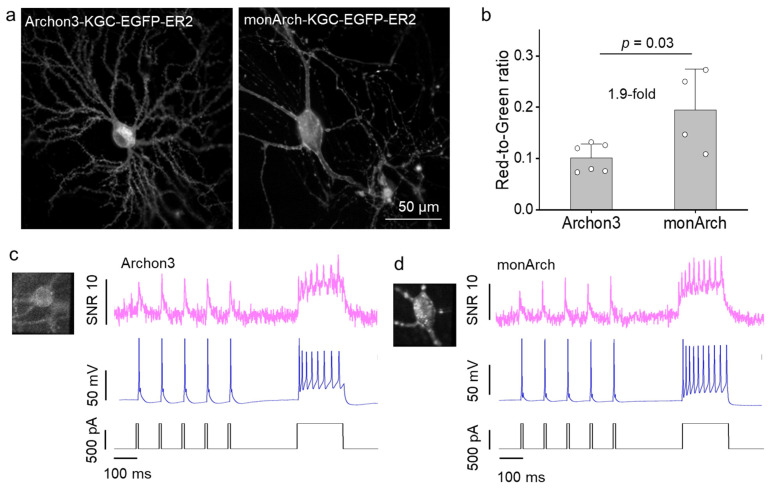
Characterization of improved Archon3-based voltage sensors in primary cultured hippocampal neurons. (**a**) Representative fluorescence images of Archon3 and monArch visualized via EGFP (excitation at 475/34BP from an LED and emission at 527/50 nm). Scale bar represents 50 µm. (**b**) Normalized fluorescent intensity of voltage sensors Archon3 and monArch. The intensity of the voltage sensor was normalized to the green GFP signal. Excitation 637 nm laser light at 1.5 W/mm^2^ and emission at 664 nm long pass. *n* = 6 individual neurons for Archon3 and *n* = 4 individual neurons for monArch. Data are represented as means ± S.D. Two-tailed Student’s *t*-test. (**c**,**d**) Single-trial optical recordings of (**c**) Archon3 and (**d**) monArch fluorescence responses (signal-to-noise ratio, SNR) during evoked action potential via whole-cell patch clamp.

**Figure 3 ijms-26-01442-f003:**
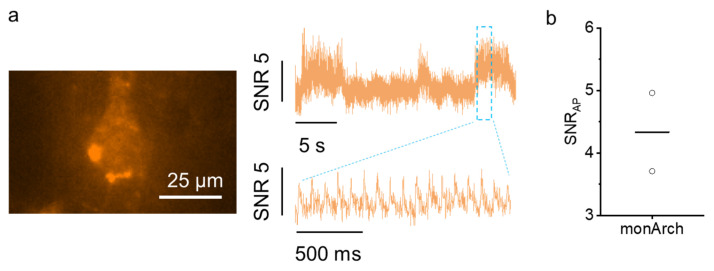
Characterization of monArch in acute mouse brain slice. (**a**) Fluorescence image of cortical neuron expressing monArch (left) and single-trial optical voltage traces (right) showing the SNR from cell on the left. (**b**) The SNR per action potential across two recordings from two neurons in one brain slice. Excitation 637 nm laser light at 1.5 W/mm^2^ and emission at 664 nm long pass. *n* = 2 neurons from one slice. The horizontal line indicates the mean value.

**Figure 4 ijms-26-01442-f004:**
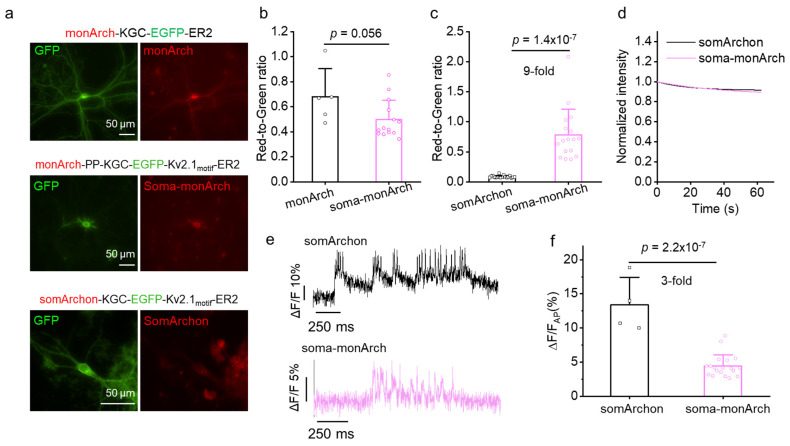
Characterization of soma-localized voltage sensor in cultured hippocampal neurons. (**a**) Representative images showing the expression of voltage sensors (red) and EGFP (green) in cultured hippocampal neurons. (**b**,**c**) Normalized fluorescent intensity of voltage sensors (**b**) monArch versus soma-monArch and (**c**) soma-monArch versus somArchon. The fluorescent intensity of the voltage sensors was normalized to the green GFP signal. Two-tailed Student’s *t*-test was used. (**d**) Photobleaching curves for somArchon and soma-monArch under continuous illumination (*n* = 4 and 7 neurons from same neuronal culture, respectively). Imaging acquisition rate: 852 Hz. somArchon and soma-monArch retain 93% and 89% of fluorescence after 60 s of continuous illumination, respectively. (**e**) Representative fluorescence optical traces of somArchon and soma-monArch. (**f**) Population data of ΔF/F for somArchon and Soma-monArch. *n* = 4 neurons from one culture for somArchon; *n* = 19 from 2 cultures for soma-monArch. Two-tailed Student’s *t*-test.

## Data Availability

All essential raw datasets including Appendix A and raw unprocessed images will be available at FigShare. The rest of the files are available from the corresponding author upon request. All plasmids used in this study are available at WeKwikGene (https://wekwikgene.wllsb.edu.cn/, accessed on 4 February 2025). Sequences of the gene will be deposited in GenBank. Source data files are provided with this paper.

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
