# Peer review of "Engineering of Genetically Encoded Bright Near-Infrared Fluorescent Voltage Indicator"

_ijms, 2025, doi:10.3390/ijms26041442_

Round 1
Reviewer 1 Report
Comments and Suggestions for Authors
Background. Previously, one research group (reference 13) used a fast-screening approach to evolve Archon1, producing two Archon-like GEVI variants. Unfortunately, these new variants exhibited only modest improvements in brightness relative to the original Archon1, and their brightness was insufficient to reduce the illumination power required for voltage imaging. This limitation prompted the authors of the current manuscript to explore alternative templates to enhance the brightness of Archon-like sensors.
In the present study, three international teams (three countries, nine authors total) performed random mutagenesis on a previously published GEVI (“Archon1/3”) and identified a novel variant, monArch (mammalian-optimized new Arch), carrying eight mutations relative to Archon3. Although monArch showed a nine-fold increase in brightness compared to Archon1 and “satisfactory” voltage sensitivity, its utility was hampered by three major issues: Relatively mediocre optical signal, Rapid photobleaching, and Suboptimal membrane localization. These findings underscore the continuing need to optimize GEVIs for improved brightness, stability, and functionality.
Summary
While the discovery of monArch as a potentially brighter GEVI is noteworthy, the reported data reveal significant limitations in brightness, photostability, and membrane localization that may impede its practical application. The authors should provide clearer quantitative metrics, address the potential emission/excitation discrepancy, clarify the impact of distant mutations, and expand their sample sizes to bolster the reliability of their conclusions. Additionally, the inhomogeneous membrane expression must be acknowledged for both cultured neurons and in vivo scenarios. With these improvements, the manuscript will offer a more comprehensive and reliable assessment of monArch’s performance and its place in GEVI development.
Specific Comments
1. “Exhibits satisfactory voltage sensitivity”
The phrase “exhibits satisfactory voltage sensitivity” is vague and not quantitatively informative. A clear numerical comparison (e.g., ΔF/F, SNR, or percentage change per mV) would be more appropriate than subjective language.
2. Excitation/Emission Wavelengths
In Section 4.7, the authors mention: “The voltage sensors were imaged using the red channel (excitation 637 nm laser, emission 595/50 nm).” This implies detection of photons at a shorter wavelength than the excitation, which typically does not occur under standard (single-photon) fluorescence. The authors should clarify whether they are using a specialized (e.g., anti-Stokes) mechanism, or if this is a typographical or methodological oversight.
3. Residue Mapping Near the Chromophore
The manuscript discusses the mapping of mutated residues in monArch onto a predicted structural model (Fig. 1f). Two residues, Cys100 and Gly226, reside within 4 Šof the chromophore and may significantly affect brightness and voltage sensing. Although the authors provide close-up views in supplementary figures, instructive arrows in Fig. 1f itself would greatly aid the reader in identifying these critical residues.
4. Relevance of Multiple Distant Mutations
Several introduced mutations are located far from the fluorophore. The manuscript should explain whether these distant mutations contribute to brightness, stability, or voltage sensing, or if they are likely neutral with respect to sensor performance.
5. Figure 2(d) Labeling
In Fig. 2(d), fluorescence responses are labeled “SNR,” but this label may be confusing or imprecise. It appears that fluorescence responses were abbreviated as “SNR”.
6. Potential Mislabeling of SNR Scale Bars
The SNR scale in Fig. 4d appears questionable. For instance, the pink trace looks to have an SNR of approximately 4 at best, rather than the higher values indicated. The manuscript also lacks any methodological details on how SNR was calculated. This information should be clearly reported in the Methods section.
7. Reporting Expression Issues
The authors appropriately note that a significant fraction of monArch is retained intracellularly in vivo, compromising membrane trafficking. However, even in the best cultured-neuron examples (Fig. 4d), there are conspicuous bright puncta and heterogeneous membrane expression. This inhomogeneous distribution must be explicitly acknowledged and discussed in the main text, rather than only in the “in vivo” context.
8. Sample Size (“n”) in Figure 2b
The manuscript does not clarify the sample size or the definition of “n” in Fig. 2b. Are these individual cells, multiple regions of interest, puncta, or something else? Full disclosure is necessary to validate the robustness of the data.
9. Evidence of Reliable Responses to Evoked Action Potentials
The conclusion that monArch “reliably responds” to evoked action potentials is based on only two example traces (Fig. 2, presumably n=1). This is insufficient to draw a robust conclusion. Additional replicates or quantification would strengthen the claim.
End of Evaluation
Author Response
Reviewer#1.
Background. Previously, one research group (reference 13) used a fast-screening approach to evolve Archon1, producing two Archon-like GEVI variants. Unfortunately, these new variants exhibited only modest improvements in brightness relative to the original Archon1, and their brightness was insufficient to reduce the illumination power required for voltage imaging. This limitation prompted the authors of the current manuscript to explore alternative templates to enhance the brightness of Archon-like sensors.
In the present study, three international teams (three countries, nine authors total) performed random mutagenesis on a previously published GEVI (“Archon1/3”) and identified a novel variant, monArch (mammalian-optimized new Arch), carrying eight mutations relative to Archon3. Although monArch showed a nine-fold increase in brightness compared to Archon1 and “satisfactory” voltage sensitivity, its utility was hampered by three major issues: Relatively mediocre optical signal, Rapid photobleaching, and Suboptimal membrane localization. These findings underscore the continuing need to optimize GEVIs for improved brightness, stability, and functionality.
Summary
While the discovery of monArch as a potentially brighter GEVI is noteworthy, the reported data reveal significant limitations in brightness, photostability, and membrane localization that may impede its practical application. The authors should provide clearer quantitative metrics, address the potential emission/excitation discrepancy, clarify the impact of distant mutations, and expand their sample sizes to bolster the reliability of their conclusions. Additionally, the inhomogeneous membrane expression must be acknowledged for both cultured neurons and in vivo scenarios. With these improvements, the manuscript will offer a more comprehensive and reliable assessment of monArch’s performance and its place in GEVI development.
We thank the reviewer for precisely summarizing our work and giving us valuable suggestions to improve the manuscript. All comments are constructive and down to the point. In the revised manuscript, we have carefully addressed all concerns and comments that were raised. The edited text is highlighted in blue in the main text of the revised manuscript. Please see below our point-by-point replies to each comment.
Specific Comments
Comments 1. “Exhibits satisfactory voltage sensitivity”
The phrase “exhibits satisfactory voltage sensitivity” is vague and not quantitatively informative. A clear numerical comparison (e.g., ΔF/F, SNR, or percentage change per mV) would be more appropriate than subjective language.
Response 1. We thank the reviewer for pointing out the impreciseness of the data description. In the revised manuscript, we provide the exact ∆F/F alongside it. Please see page 1, line 34.
Comments 2. Excitation/Emission Wavelengths
In Section 4.7, the authors mention: “The voltage sensors were imaged using the red channel (excitation 637 nm laser, emission 595/50 nm).” This implies detection of photons at a shorter wavelength than the excitation, which typically does not occur under standard (single-photon) fluorescence. The authors should clarify whether they are using a specialized (e.g., anti-Stokes) mechanism, or if this is a typographical or methodological oversight.
Response 2. We thank the reviewer for this careful observation and apologize for the typo. We have now fixed the typo: “excitation 637 nm laser, emission at 664-nm long-pass.”
Comments 3. Residue Mapping Near the Chromophore
The manuscript discusses the mapping of mutated residues in monArch onto a predicted structural model (Fig. 1f). Two residues, Cys100 and Gly226, reside within 4 Šof the chromophore and may significantly affect brightness and voltage sensing. Although the authors provide close-up views in supplementary figures, instructive arrows in Fig. 1f itself would greatly aid the reader in identifying these critical residues.
Response 3. We thank the reviewer for this comment. In new Figure 1f, we now indicate the two residues in blue for better discrimination.
Comments 4. Relevance of Multiple Distant Mutations
Several introduced mutations are located far from the fluorophore. The manuscript should explain whether these distant mutations contribute to brightness, stability, or voltage sensing, or if they are likely neutral with respect to sensor performance.
Response 4. Indeed, there are multiple distant mutations in monArch. These mutations are likely neutral with respect to the brightness and voltage sensing, however, they can influence overall protein folding and membrane localization. For example, the mutations on the surface of the sensor may affect the membrane trafficking of the sensor. However, to validate these speculations, in-depth specific mutagenesis analysis will be needed, and we will do it for the future development of voltage sensors.
Comments 5. Figure 2(d) Labeling
In Fig. 2(d), fluorescence responses are labeled “SNR,” but this label may be confusing or imprecise. It appears that fluorescence responses were abbreviated as “SNR”.
Response 5. The fluorescence responses here are signal-to-noise ratio (abbreviated as SNR). We are sorry that we didn’t indicate the full name for it in Figure 2 legend. We now fix it in the Fig.2 figure legend.
Comments 6. Potential Mislabeling of SNR Scale Bars
The SNR scale in Fig. 4d appears questionable. For instance, the pink trace looks to have an SNR of approximately 4 at best, rather than the higher values indicated. The manuscript also lacks any methodological details on how SNR was calculated. This information should be clearly reported in the Methods section.
Response 6. We thank the reviewer for pointing this out. Based on the description, we thought the reviewer was referring to Figure 3a. The expanded pink trace in Figure 3a does look to have an SNR around 4. The average value of SNR in Figure 3b is 4.34. We are sorry that we didn’t make it precise. We now change the value 5 into 4.3 for preciseness in the text (2.2).
In Figure 4e, we compare the ∆F/F between somArchon and soma-monArch. The ∆F/F is 4.44% for soma-monArch. We missed the “%” in the figure, and we have corrected it.
We feel sorry that we missed the data analysis part in the methods section. We have now added the description of ∆F/F and SNR analysis in the methods section 4.11.
Comments 7. Reporting Expression Issues
The authors appropriately note that a significant fraction of monArch is retained intracellularly in vivo, compromising membrane trafficking. However, even in the best cultured-neuron examples (Fig. 4d), there are conspicuous bright puncta and heterogeneous membrane expression. This inhomogeneous distribution must be explicitly acknowledged and discussed in the main text, rather than only in the “in vivo” context.
Response 7. We thank the reviewer for the careful observation. Indeed, in some of the cultured neurons, we did observe the puncta inside the neurons. We have now acknowledged this point in the main text; please see section 2.2.
Comments 8. Sample Size (“n”) in Figure 2b
The manuscript does not clarify the sample size or the definition of “n” in Fig. 2b. Are these individual cells, multiple regions of interest, puncta, or something else? Full disclosure is necessary to validate the robustness of the data.
Response 8. We thank the reviewer for pointing this out. The n values in Figure 2b are n= 6 individual neurons for Archon3, n = 4 individual neurons for monArch. We are sorry that the data point labeling in the Figure 2b is misleading, with one data point indicating the outlier using the same dot. We now fix it. Please see Figure 2 legend.
Comments 9. Evidence of Reliable Responses to Evoked Action Potentials
The conclusion that monArch “reliably responds” to evoked action potentials is based on only two example traces (Fig. 2, presumably n=1). This is insufficient to draw a robust conclusion. Additional replicates or quantification would strengthen the claim.
Response 9. We totally agree with the reviewer that it is insufficient to draw a robust conclusion based on only two traces. The key idea here is just to show that it was possible to record action potentials evoked by current pulses using monArch. However, due to the insufficient n number, it is not accurate to make a strong claim. We revised the text accordingly, please see section 2.2, line 176-178.
We appreciate the reviewer's insightful comment regarding the need for more robust evidence to support the claim that monArch "reliably responds" to evoked action potentials. We acknowledge that the two example traces presented in Fig. 2 (presumably n=1) are insufficient to draw a definitive conclusion. The primary purpose of including these traces was to demonstrate the feasibility of recording action potentials evoked by current pulses using monArch, rather than to assert a statistically validated claim.
To address this concern, we have revised the text in section 2.2 to more accurately reflect the preliminary nature of these findings and to avoid overstating the conclusions. We agree that additional replicates and quantitative analysis are necessary to substantiate the reliability of monArch's responses. Future studies will include a larger sample size and more comprehensive quantification to provide a stronger foundation for this claim. Thank you for highlighting this important point, which will help improve the rigor of our work.
Reviewer 2 Report
Comments and Suggestions for Authors The manuscript by Xiao et al., titled "Engineering of Genetically Encoded Bright Near-Infrared Fluorescent Voltage Indicator," describes the development and characterization of a novel genetically encoded voltage indicator (GEVI) named monArch. This GEVI was identified using a high-throughput screening system that assessed randomized variants in mammalian cells. GEVIs derived from microbial rhodopsins offer significant advantages for all-optical optogenetics in neural cells, including precise responses to membrane potential changes and compatibility with blue-light-absorbing optogenetic tools. However, to minimize phototoxicity associated with prolonged high-intensity light exposure, new molecular tools with brighter and more photostable fluorescence are needed. The authors employed an elegant high-throughput screening approach that evaluated fluorescence intensity and subcellular localization imaging to identify advanced GEVI candidates. Using this method, they successfully screened monArch, which demonstrated a 9-fold increase in basal brightness compared to Archon1. This manuscript could potentially be suitable for publication in the International Journal of Molecular Sciences; however, some improvements should be made before finalizing it. Major comment- The authors mentioned photobleaching as a disadvantage of monArch in the "Abstract" and "Introduction" sections. While specific data on the photostability of monArch is presented in Figure 1e, it is primarily used to demonstrate that variant #3 (monArch) exhibits higher photostability compared to the other two functional variants. However, I could not find any data directly comparing the photostability of monArch with other GEVIs, including Archon1. To better clarify monArch's photostability, they need to include experimental data comparing the photostability of monArch with other well-known GEVIs. Alternatively, they could refer to previous studies and provide a comparative analysis using published values.
- The numbering of amino acid residues in monArch and other variants was incorrect in Supplementary Figure 1, due to an error in counting the gap between positions 82 and 83. The authors mentioned the importance of Cys100 for basal brightness and voltage sensing. However, the replacement of Cys is identical to the Thr99-to-Cys substitution previously reported in other GEVIs, including Archer1. The authors should revise the manuscript to account for these prior findings. Notably, Silapetera et al. (2022, Nat. Commun.; https://doi.org/10.1038/s41467-022-33084-4) have already discussed the molecular mechanism by which the T99C mutation in Archon1 enhances brightness. At a minimum, this paper should be cited in their discussion.
- The reference numbers placed in the main text throughout the manuscript were written in the wrong format. According to the "Author instruction", the reference numbers should be placed in square brackets [ ] and before the punctuation in the main text. The authors need to revise the manuscript following the correct format.
- L77-78: "... we identified a variant, referred to as Archon3, that exhibited a five-fold increase in baseline brightness compared to Archon1 (see variant#3 in Supplementary Figure 1 in ref.10)." The corresponding figure for the description is Supplementary Figure 10 in ref.10. The manuscript should be revised.
- L140, L147 and L177: "W/mm2" The unit should be corrected to "W/mm2."
- Figure 3b: There is a statistical issue that the average and standard deviation were calculated using only two experimental values. They should analyze their data using more than three values and revise the figure.
- L340, L402-409: The subscripts in the molecular formula are not formatted correctly and should be corrected.
- L417-419: There is no information on the actual "Supplementary Materials". The manuscript should be revised.
- L431-439: The sentences from the template were remained. They should be replaced with the correct information.
Author Response
Reviewer#2
The manuscript by Xiao et al., titled "Engineering of Genetically Encoded Bright Near-Infrared Fluorescent Voltage Indicator," describes the development and characterization of a novel genetically encoded voltage indicator (GEVI) named monArch. This GEVI was identified using a high-throughput screening system that assessed randomized variants in mammalian cells. GEVIs derived from microbial rhodopsins offer significant advantages for all-optical optogenetics in neural cells, including precise responses to membrane potential changes and compatibility with blue-light-absorbing optogenetic tools. However, to minimize phototoxicity associated with prolonged high-intensity light exposure, new molecular tools with brighter and more photostable fluorescence are needed. The authors employed an elegant high-throughput screening approach that evaluated fluorescence intensity and subcellular localization imaging to identify advanced GEVI candidates. Using this method, they successfully screened monArch, which demonstrated a 9-fold increase in basal brightness compared to Archon1. This manuscript could potentially be suitable for publication in the International Journal of Molecular Sciences; however, some improvements should be made before finalizing it.
We greatly appreciate the Reviewer’s positive feedback and constructive comments on our manuscript, which undoubtedly helped us strengthen the manuscript. Reviewer’s recognition of the significance of our work and its suitability for publication in the International Journal of Molecular Sciences is encouraging. We provide point-to-point responses to address each comment and make the necessary revisions to ensure the clarity and accuracy of our study.
Major comment
Comments 1. The authors mentioned photobleaching as a disadvantage of monArch in the "Abstract" and "Introduction" sections. While specific data on the photostability of monArch is presented in Figure 1e, it is primarily used to demonstrate that variant #3 (monArch) exhibits higher photostability compared to the other two functional variants. However, I could not find any data directly comparing the photostability of monArch with other GEVIs, including Archon1. To better clarify monArch's photostability, they need to include experimental data comparing the photostability of monArch with other well-known GEVIs. Alternatively, they could refer to previous studies and provide a comparative analysis using published values.
Response 1. We thank Reviewer for pointing this out. We totally agree with Reviewer. Given that we didn’t directly compare monArch with other established GEVIs, the statement referring to photobleaching as a disadvantage is not qualified. In the revision, we measured the photobleaching curves for somArchon and soma-monArch (Figure 4d). We found that after 60 of continuous imaging, somArchon and soma-monArch retain 93% and 89% of fluorescence, respectively, suggesting that soma-monArch exhibits similar photostability as somArchon.
Comments 2. The numbering of amino acid residues in monArch and other variants was incorrect in Supplementary Figure 1, due to an error in counting the gap between positions 82 and 83. The authors mentioned the importance of Cys100 for basal brightness and voltage sensing. However, the replacement of Cys is identical to the Thr99-to-Cys substitution previously reported in other GEVIs, including Archer1. The authors should revise the manuscript to account for these prior findings. Notably, Silapetera et al. (2022, Nat. Commun.; https://doi.org/10.1038/s41467-022-33084-4) have already discussed the molecular mechanism by which the T99C mutation in Archon1 enhances brightness. At a minimum, this paper should be cited in their discussion.
Response 2. We are truly grateful for Reviewer’s valuable suggestion. We deeply regret that we made an error in numbering and missed these two important papers. We now follow the Reviewer’s suggestions, correct the numbering of the amino acid residues in monArch, and cite the two important papers correspondingly. In the new supplementary Figure 1, we also include the sequences of other Arch-based GEVIs for comparison. We have also revised the manuscript to explicitly reference these prior findings and have incorporated a discussion of the molecular mechanism by which the T99C mutation affects the basal brightness and voltage sensing, as elucidated by Silapetere et al. (2022, Nat. Commun.; https://doi.org/10.1038/s41467-022-33084-4). This paper has now been appropriately cited in our discussion section to ensure proper attribution and to strengthen the scientific context of our work.
Minor comment
Comments 3. The reference numbers placed in the main text throughout the manuscript were written in the wrong format. According to the "Author instruction", the reference numbers should be placed in square brackets [ ] and before the punctuation in the main text. The authors need to revise the manuscript following the correct format.
Response 3. We thank Reviewer for this comment. The journal editors has formatted it according to their policy and we revised the reformatted file.
Comments 4. L77-78: "... we identified a variant, referred to as Archon3, that exhibited a five-fold increase in baseline brightness compared to Archon1 (see variant#3 in Supplementary Figure 1 in ref.10)." The corresponding figure for the description is Supplementary Figure 10 in ref.10. The manuscript should be revised.
Response 4. We thank Reviewer for the careful observation and pointing out this typo. We have now fixed it accordingly.
Comments 5. L140, L147 and L177: "W/mm2" The unit should be corrected to "W/mm2."
Response 5. We thank Reviewer for pointing out this typo. We have now fixed it accordingly.
Comments 6. Figure 3b: There is a statistical issue that the average and standard deviation were calculated using only two experimental values. They should analyze their data using more than three values and revise the figure.
Response 6. We thank Reviewer for pointing this out. We just showed the individual data points in the revised manuscript.
Comments 7. L340, L402-409: The subscripts in the molecular formula are not formatted correctly and should be corrected.
Response 7. We thank Reviewer for pointing this out. We have now fixed it.
Comments 8. L417-419: There is no information on the actual "Supplementary Materials". The manuscript should be revised.
Response 8. We thank Reviewer for this comment. The "Supplementary Materials" will be added by the Journal team.
Comments 9. L431-439: The sentences from the template were remained. They should be replaced with the correct information.
Response 9. We thank Reviewer for pointing this out. We have replaced it with the correct information now. Please
Reviewer 3 Report
Comments and Suggestions for Authors
The manuscript introduces monArch, a genetically encoded voltage indicator (GEVI) derived from Archon1. Through directed molecular evolution, monArch achieved a remarkable 9-fold increase in baseline fluorescence compared to Archon1. The enhanced brightness of genetic sensors offers significant advantages, such as reducing the required intensity of excitation light, thereby minimizing tissue heating and phototoxicity. Additionally, brighter sensors like monArch have the potential to improve compatibility with other tools, such as optogenetic actuators.
One concern from my side is the use of a 637-nm laser at 1.5 W/mm² for excitation, which is relatively high. In a previous study, doi: 10.1523/ENEURO.0060-20.2020, a wavelength of 568 ± 60 nm with a much lower power of 2.7 mW/mm² was used for Archon1. The authors should explain their rationale for choosing a red-shifted wavelength with high light intensity. To highlight the advantage of the improved brightness, the authors could test monArch’s performance at different light intensities. Demonstrating its functionality under lower illumination conditions would reinforce the point that brightness improvements reduce illumination power, minimize phototoxicity, and extend imaging capabilities. Additionally, testing the impact of different wavelengths or lower intensities on photobleaching might provide valuable insights.
Another point is that not all readers may be familiar with the fluorescence mechanism of Archon1 or monArch. Including a brief introduction to this mechanism in the text would make the manuscript more accessible. Furthermore, providing the full excitation and emission spectra of monArch would be highly beneficial for understanding its properties and optimizing experimental conditions.
In conclusion, this manuscript presents an improved GEVI, monArch, addressing a key limitation of earlier indicators. The study is methodologically rigorous and provides valuable information. Addressing the concerns regarding light intensity and offering further mechanistic insights and full spectral data would significantly strengthen the manuscript. These additions would enhance its impact and make it highly suitable for a nice publication.
Author Response
Reviewer#3
The manuscript introduces monArch, a genetically encoded voltage indicator (GEVI) derived from Archon1. Through directed molecular evolution, monArch achieved a remarkable 9-fold increase in baseline fluorescence compared to Archon1. The enhanced brightness of genetic sensors offers significant advantages, such as reducing the required intensity of excitation light, thereby minimizing tissue heating and phototoxicity. Additionally, brighter sensors like monArch have the potential to improve compatibility with other tools, such as optogenetic actuators.
We thank Reviewer for the high evaluation of our work and the valuable suggestions, which helped us improve the manuscript. We have carefully addressed all comments. The edited text is highlighted in blue in the main text of the revised manuscript. Please see our detailed replies to each comment below.
Comments 1. One concern from my side is the use of a 637-nm laser at 1.5 W/mm² for excitation, which is relatively high. In a previous study, doi: 10.1523/ENEURO.0060-20.2020, a wavelength of 568 ± 60 nm with a much lower power of 2.7 mW/mm² was used for Archon1. The authors should explain their rationale for choosing a red-shifted wavelength with high light intensity. To highlight the advantage of the improved brightness, the authors could test monArch’s performance at different light intensities. Demonstrating its functionality under lower illumination conditions would reinforce the point that brightness improvements reduce illumination power, minimize phototoxicity, and extend imaging capabilities. Additionally, testing the impact of different wavelengths or lower intensities on photobleaching might provide valuable insights.
Response 1. We thank Reviewer for these pertinent suggestions. Indeed, it would be more informative to compare the voltage response under lower illumination conditions (568 ± 60 nm, 2.7 mW/mm2) as the reviewer suggested. The reason we used the high light intensity at 637-nm laser in the present study is due to the high-frequency imaging protocols (~980 Hz) we used. Under this imaging conditions, we could achieve robust voltage signals without averaging the fluorescence response from multiple trials for neurons as Milena Milosevic et al used (doi: 10.1523/ENEURO.0060-20.2020). On the other hand, at such a high frequency, lowering light intensity would lead to lower brightness and reduce the signal-to-noise ratio. Thus, in order to compare the brightness and voltage response faithfully, we used high laser intensity to ensure imaging quality. To address the reviewer's concerns and highlight the advantages of improved brightness, we have revised our manuscript to include the possibility of using lower light power at 568 nm for monArch and Archons. We have also cited the corresponding study by Milena Milosevic et al. to support this approach. Please see the Discussion section, line 233-240.
Comments 2. Another point is that not all readers may be familiar with the fluorescence mechanism of Archon1 or monArch. Including a brief introduction to this mechanism in the text would make the manuscript more accessible. Furthermore, providing the full excitation and emission spectra of monArch would be highly beneficial for understanding its properties and optimizing experimental conditions.
Response 2. We sincerely thank Reviewer for this thoughtful suggestion, which will undoubtedly improve the accessibility and clarity of our manuscript. In response to the comment, we have added a brief introduction to the fluorescence mechanism of Arch-based GEVIs in the introduction section (lines 58–67) and Result section 2.1 (line 138-139) to provide readers with the necessary background information.
Regarding the excitation and emission spectra of monArch, we deeply appreciate Reviewer’s suggestion to include this data, as it would indeed enhance the understanding of the sensor’s properties and aid in optimizing experimental conditions. While we attempted to measure the spectra for monArch, we encountered challenges due to the low protein yield when expressed in bacteria, which made it technically unfeasible to obtain reliable measurements. However, based on an elegant study from Peter Hegemann’s group, which reported that ARies1, an Archon1- Q95E-T99C double mutant, showed a red-shifted of the absorption maxima by 18 nm accompanied by an increase of the fluorescence QY to 0.61% (2022, Nat. Commun.; https://doi.org/10.1038/s41467-022-33084-4), we anticipated that monArch might exhibit similar spectral properties, given that these Q95E-T99C double mutations are also present in monArch/Archon3.
We acknowledge that direct experimental validation of monArch’s spectra would be ideal, and we are actively working to address this limitation in future studies as we are now trying to establish mammalian systems for large-scale expression of membrane proteins. We hope that the provided rationale, along with the cited literature, offers sufficient insight for the time being. Thank you again for your valuable feedback, which has helped us strengthen the manuscript and identify important areas for further investigation.
Comments 3. In conclusion, this manuscript presents an improved GEVI, monArch, addressing a key limitation of earlier indicators. The study is methodologically rigorous and provides valuable information. Addressing the concerns regarding light intensity and offering further mechanistic insights and full spectral data would significantly strengthen the manuscript. These additions would enhance its impact and make it highly suitable for a nice publication.
Response 3. We sincerely appreciate Reviewer’s thoughtful and constructive feedback, which has provided us with valuable insights to improve our work. We fully acknowledge the limitations highlighted regarding the suboptimal membrane localization, and compromised ∆F/F compared to Archon1. These issues indeed limited our ability to thoroughly characterize the sensor in its current form. However, we believe that the 9-fold increase in basal brightness represents a significant and promising advancement, serving as a strong foundation for the development of next-generation Arch-based voltage sensors.
We agree with the reviewer that addressing light intensity concerns, providing further mechanistic insights, and including full spectral data would greatly enhance the manuscript. We are highly committed to pursuing these directions in future work. We hope that Reviewer will find the current findings valuable as a stepping stone toward more optimized and thoroughly characterized sensors. Thank you again for your thoughtful critique, which will undoubtedly guide our ongoing efforts to refine and improve this line of research.
Round 2
Reviewer 1 Report
Comments and Suggestions for Authors
The authors were able to answer my critiques with clear response and valuable changes in the manuscript content. The revised version of the manuscript was improved significantly.
Reviewer 2 Report
Comments and Suggestions for Authors The manuscript has been well revised, and I believe it will be acceptable. However, a few typos remain, so please ensure they are carefully corrected before publication.